# Research Progress of Indole Alkaloids: Targeting MAP Kinase Signaling Pathways in Cancer Treatment

**DOI:** 10.3390/cancers15225311

**Published:** 2023-11-07

**Authors:** Md. Al Amin, Talha Bin Emran, Jishan Khan, Mehrukh Zehravi, Indu Sharma, Anasuya Patil, Jeetendra Kumar Gupta, D. Jeslin, Karthickeyan Krishnan, Rajib Das, Firzan Nainu, Irfan Ahmad, Polrat Wilairatana

**Affiliations:** 1Department of Pharmacy, Faculty of Allied Health Sciences, Daffodil International University, Dhaka 1207, Bangladesh; amin29-825@diu.edu.bd; 2Department of Pathology and Laboratory Medicine, Warren Alpert Medical School & Legorreta Cancer Center, Brown University, Providence, RI 02912, USA; 3Department of Pharmacy, International Islamic University Chittagong, Kumira, Chittagong 4318, Bangladesh; 4Department of Clinical Pharmacy, College of Dentistry & Pharmacy, Buraydah Private Colleges, Buraydah 51418, Saudi Arabia; ahrukh.zehravi@hotmail.com; 5Department of Physics, Career Point University, Hamirpur 176041, Himachal Pradesh, India; 6Department of Pharmaceutics, KLE College of Pharmacy, Bengaluru 560010, Karnataka, India; 7Department of Pharmacology, Institute of Pharmaceutical Research, GLA University, Mathura 281406, Uttar Pradesh, India; jk.gupta@gla.ac.in; 8Department of Pharmaceutics, Sree Balaji Medical College and Hospital Campus, Bharath Institute of Higher Education and Research, Chromepet, Chennai 600044, Tamil Nadu, India; 9Department of Pharmacy Practice, School of Pharmaceutical Sciences, Vels Institute of Science, Technology and Advanced Studies (VISTAS), Pallavaram, Chennai 600117, Tamil Nadu, India; karthickeyanpharmacy@gmail.com; 10Department of Pharmacy, Faculty of Pharmacy, University of Dhaka, Dhaka 1000, Bangladesh; 11Department of Pharmacy, Faculty of Pharmacy, Hasanuddin University, Makassar 90245, Indonesia; firzannainu@unhas.ac.id; 12Department of Clinical Laboratory Sciences, College of Applied Medical Sciences, King Khalid University, Abha 61411, Saudi Arabia; 13Department of Clinical Tropical Medicine, Faculty of Tropical Medicine, Mahidol University, Bangkok 10400, Thailand

**Keywords:** apoptosis, clinical studies, cancer, indole alkaloids, MAPK

## Abstract

**Simple Summary:**

Cancer is a major global health concern, with the Mitogen-activated protein kinase (MAPK) pathway playing a crucial role in cancerous diseases. Indole alkaloids, such as vincristine and evodiamine, have been extensively studied for their potential in cancer treatment. These naturally occurring indole alkaloids have strong anticancer properties, and research is ongoing to develop new anticancer small molecules. The current review aims to evaluate how indole alkaloids affect the MAPK signaling pathway in cancer treatment and focus on advancements in the role of indole alkaloids. The review also highlights clinical trials with indole alkaloids in cancer treatment.

**Abstract:**

Cancer is the leading cause of morbidity and mortality in people throughout the world. There are many signaling pathways associated with cancerous diseases, from which the Mitogen-activated protein kinase (MAPK) pathway performs a significant role in this regard. Apoptosis and proliferation are correlated with MAPK signaling pathways. Plenty of experimental investigations were carried out to assess the role of indole alkaloids in MAPK-mediated cancerous diseases. Previous reports established that indole alkaloids, such as vincristine and evodiamine are useful small molecules in cancer treatment via the MAPK signaling system. Indole alkaloids have the anticancer potential through different pathways. Vincristine and evodiamine are naturally occurring indole alkaloids that have strong anticancer properties. Additionally, much research is ongoing or completed with molecules belonging to this group. The current review aims to evaluate how indole alkaloids affect the MAPK signaling pathway in cancer treatment. Additionally, we focused on the advancement in the role of indole alkaloids, with the intention of modifying the MAPK signaling pathways to investigate potential new anticancer small molecules. Furthermore, clinical trials with indole alkaloids in cancer treatment are also highlighted.

## 1. Introduction

The cancer is initiated when cell division and proliferation are not properly regulated, which might lead to the disease [1]. According to GLOBOCAN, the number of cancer cases around the globe is anticipated to reach 28.4 million in the year 2040 [2]. According to the World Health Organization (WHO), smoking, drinking alcohol, not exercising, bad diet, air pollution, and even some persistent viral illnesses may lead to cancer [3]. Conventional treatment methods have been used for a long time, including chemotherapy, surgery, and radiation; however, stem cell therapy, chemodynamic, targeted therapy, sonodynamic, natural antioxidants, ablation therapy, nanoparticles, radionics, and ferroptosis-based therapies have advanced in recent years [4]. Mitogen-activated protein kinase (MAPK) cascades are critical signaling channels that regulate many biological processes, including apoptosis and proliferation [5]. Researchers have discovered that polyphenols may inhibit the growth of cancer cells via three distinct pathways: the extrinsic death-receptor route, the intrinsic mitochondrial pathway, and the apoptotic perforin-granzyme system. It has been proven that the MAPK signaling pathway has a role in the course of the cell cycle, metabolism, aging and development, reproduction, cancer, and other activities [6,7]. In several kinds of cancers, polyphenols regulate the MAPK signaling system to provide an anticancer effect by inducing apoptotic responses in the target cells. Despite this, several genomic studies have shown that the MAPK pathway is often altered in a variety of cancer cell lines, which compromises the basic function that it plays [8,9].

The MAPK signaling pathway is an essential component of the human body. It plays a role in a number of important cellular processes, including survival, proliferation, differentiation, inflammation, and apoptosis [10]. Additionally, it has an effect on the development of treatment resistance in cancer cells [11]. The extracellular signal–regulated kinase (ERK), c-jun N-terminal kinase (JNK), and the p38 kinase are the three MAPKs that are found in human cells [12]. While c-Jun N-terminal kinase is important in the inflammatory response, apoptosis, and autophagy, ERK is principally responsible for the effects that it has on cell genesis, proliferation, and differentiation. P38 kinase has been linked to the inflammatory response, apoptosis, and autophagy, and it also plays a critical role as an inhibitor in the development of cancer [13]. A stress-activated protein known as JNK takes part in the cell’s response to stress, as well as in its differentiation, proliferation, and apoptosis, but its primary function is in the death of tumor cells [14,15]. There have been major breakthroughs achieved in both drug sensitivity and drug resistance processes as a result of recent research on medicines that target the MAPK pathway for the treatment of cancer [16]. The MAPK signaling pathway is complicated, and more research has to be performed on its individual processes. As a result, anticancer medications that target the MAPK signaling pathway provide an exciting new avenue [10]. The indole alkaloids that are obtained from plant sources possess a variety of biological activities, including those that are antiparasitic, cytotoxic, antiserotonin, anti-inflammatory, and antiviral [17,18]. Vinblastine, vincristine, and camptothecin are all examples of monoterpene indole alkaloids used to treat cancer, whereas reserpine and ajmalicine are used to treat hypertension, and 7-hydroxymitragynine is used as an analgesic [19]. So, indole alkaloids could be choice of interests to treat cancers associated with the MAPK signaling pathway.

Therefore, the ultimate goal is to explore the role of the MAPK signaling pathway in cancer progression. Additionally, the anticancer effects of indole alkaloids in several cancer types via the modification of MAPK signaling pathways with their preclinical findings. Some clinical data have also been reported to validate the preclinical evidence along with subsequent advancements in drug discovery in this regard.

## 2. MAPK Signaling Pathway in Cancer

The MAPKs that have been studied the most are ERKs, JNK, and p38 MAPKs [20]. Kudaravalli et al. reported that cancer stem cells are what make cancer resistant to treatment and spread. p38 MAPK is crucial for cancer stem cell growth, metastasis, and treatment resistance [21]. Hong et al. demonstrated that, in breast cancer cells, activating the p38 pathway can play a key role in stopping tumor spreading. However, enhanced miR-365 suppression of IL-6 production is linked to p38 activation and human breast cancer metastasis [22]. The p38 MAPK protein is believed to possess substantial involvement in several cellular processes, including apoptosis, survival, differentiation, proliferation, inflammation, development, and various stress responses. The involvement of p38 activity is necessary for the G1/S cell cycle arrest caused by Cdc42. The suppression of cyclinD1 expression may serve as a mediator for the inhibitory function. The activation of p38 has been observed to induce mitotic arrest [23,24]. Recent research has shown that the protein p38 has functionality in the differentiation of several distinct types of vertebrate cells, including chondroblasts, adipocytes, myoblasts, cardiomyocytes, erythroblasts, and neurons [25]. 

A study has shown that silencing JNK1 and JNK2 with siRNA stopped WM164 melanoma cells from growing, spreading, and invading [26]. Chen et al. reported that cellular responses to various stimuli are mediated by the JNK pathway. This pathway is also essential for apoptotic signaling. Different patterns of JNK activation generated by apoptotic stimuli and other stimuli indicate that the length of JNK activation dictates the result of cellular communication mediated by this pathway [27]. The JNK pathway is vital for the apoptosis of breast cancer cells triggered by microtubule-damaging agents because it phosphorylates and inactivates the mitochondrial anti-apoptotic protein Bcl2 [28,29]. The activation of JNKs is linked to transformation across a wide variety of oncogene- and growth factor-mediated systems. It is believed that an essential part of this procedure is played by the transactivation of c-Jun. JNKs are capable of transducing signals that are necessary for differentiating in the hematological system and may also have a role in the development of embryos. There is evidence that the JNK pathway is involved in both apoptosis and survival signaling. It has been observed that UV exposure causes fibroblasts to undergo apoptosis, which in turn needs JNK in order for cytochrome C to be released from the mitochondria [24,30,31]. 

Kohno et al. reported that, by inhibiting the constitutively active ERK pathway, upstream MEK inhibitors sensitize tumor cells to cytotoxic cancer therapies-induced apoptosis [32]. The ERK pathway performs a significant role in regulating various cellular biological processes, including the proliferation of cells, cell cycle control, cell division, cell apoptosis, and tissue formation. Additionally, there is a significant association between this pathway and tumor formation, as evidenced by previous studies [5,33]. The presence of elevated ERK expression has been observed in different types of human tumors, including breast, colon, ovarian, and lung cancer [34,35,36]. 

## 3. Indole Alkaloids Targeting MAPK in Cancers 

### 3.1. Evodiamine

Du et al. reported that the expression of p-ERK and p-p38 MAPK are regulated by evodiamine. Evodiamine-associated apoptosis was increased by ERK inhibitor PD98059 or p38 MAPK inhibitor SB203580 [37]. ERK2 activation confers the capacity to seed and colonize the liver on colon cancer cells, whereas diminished p38 MAPK signaling confers the capacity to produce lung metastases from formerly existing liver lesions. The p38 MAPK signaling downregulation boosted the production of the cytokine PTHLH, and this cytokine disrupts colon cancer cells by causing cell death in lung microvasculature endothelial cells [38]. Chien et al. reported that JNK and ERK protein phosphorylation occurred in HT-29 and COLO205 cells treated with evodiamine. SP600125, a JNK inhibitor, reduced evodiamine-associated phosphorylated JNK production, apoptosis, and G2/M arrest in colon cancer cells [39].

A study was conducted to examine the impact of evodiamine on the growth, programmed cell death, and self-degradation of human PC cells. The study involved treating two types of PC cell lines, SW1990 and PANC-1, with varying concentrations of Evodiamine. CCK-8 assay was used to measure the cell proliferation. The study found that evodiamine dose-dependently impacts the phosphorylation of serine/threonine kinase (AKT), ERK1/2, and P38 MAPK. The experimental findings suggest that evodiamine has the ability to reduce the levels of LC3II and increase the levels of P62. The group of patients treated with evodiamine exhibited a more substantial drop in tumor weight in comparison to the control one. In vivo, the immunohistochemical detection of phosphorylated AKT tumor expression exhibited a reduction in response to escalating doses of evodiamine. The apoptosis induction in PC cells by evodiamine was observed through the inhibition of AKT and MAPK/ERK, as well as the suppression of signal transducer and phosphorylation in PC cells, which consequently led to the inhibition of autophagy. These findings suggest that evodiamine holds potential as a novel treatment for PC [40].

It has been reported by Lijuan et al. that evodiamine inhibited Akt, PI3K, ERK1/2 MAPK, and p38 MAPK, targeting the MAPK, and PI3K/Akt pathways. Evodiamine was administered to HO-8910PM cancer cells at 0, 1.25, 2.5, and 5 μM for 1–4 days. The (3-[4,5-dimethylthiazol-2-yl]-2,5 diphenyl tetrazolium bromide) (MTT) assay measured the growth of the HO-8910PM cell which was treated with evodiamine [41]. A study conducted by Zhang and colleagues revealed that evodiamine displayed a dose- and time-dependent cytotoxic action on HeLa and A375-S2 cells. The activation of Caspase-3 and -8 was observed in the apoptosis that was induced by the administration of evodiamine at a concentration of 15 micromol/L. After a 24-h incubation period of A375-S2 cells, the administration of evodiamine at a concentration of 15 micromol/L was found to induce necrosis that was associated with the activities of p38 and ERK. The cytotoxicity of evodiamine on HeLa cells was observed to be associated with cell cycle arrest at the G2/M phase. However, no significant impact of evodiamine on the cell cycle of A375-S2 cells was detected. The compound evodiamine has been observed to prompt apoptosis in HeLa cells through caspase-3 and the activation of -8 activation. This apoptotic response is associated with the G2/M phase arrest of the cell cycle. Conversely, within A375-S2 cells, evodiamine triggers caspase-3,8-mediated apoptosis during initial phases and stimulates MAPK-mediated necrosis during subsequent stages of cellular cultivation [42]. Shi et al. reported that human urothelial carcinoma cells treated with evodiamine show cell growth inhibition, carcinogenesis suppression, and apoptosis induction. Evodiamine’s effects on MAPK signaling were examined using western blotting after 5637 and HT1197 cells were cultured with the chemical. After treatment with evodiamine, P38 and JNK phosphorylation levels in both cell lines were found to be markedly elevated; HT1197 cells had enhanced ERK phosphorylation after 48 h, however 5637 cells did not [43].

A study demonstrated that evodiamine triggered apoptosis in glioma cells in such a way that was related to the dose and duration of exposure. The inhibition of calcium channels located in the endoplasmic reticulum had a notable impact on the increase in cytosolic calcium levels, apoptosis, and mitochondrial depolarization induced by evodiamine. This finding indicates that evodiamine triggers an intrinsic apoptosis pathway that is mediated by calcium. It is noteworthy that evodiamine exhibited an augmentation of autophagy, which had attained a saturation point after 24 h. The pharmacological suppression of autophagy results in a rise in apoptosis and a decline in cell viability. The induction of autophagy by evodiamine was significantly reduced upon inhibition of ER activation. The suppression of autophagy associated with an increase in apoptosis was observed upon the deactivation of c-JNK in the presence of evodiamine. The findings suggest that the stimulation of JNK may play a role in Evo-induced autophagy, as evidenced by the observed decrease in autophagy and concurrent rise in the death in cells that were pre-treated with the JNK inhibitor, SP600125 [44]. According to Hong et al., ERK phosphorylation and expression were reduced by evodiamine, whereas p38 MAPK and JNK1/2 were unaffected, indicating that in evodiamine-induced apoptosis, p38 and JNK1/2 activation were not crucial [45].

### 3.2. Vinorelbine

Liem et al. reported the fact that doxorubicin acts through the expression of p53 and vinorelbine activates p38, which may explain the high clinical response rate when the two drugs are used to treat breast cancer [46].

### 3.3. Hirsutine

The compound hirsutine (Figure 1) exhibited significant cytotoxic effects on the BT474 and MDA-MB 453, which are known to be p53-mutated and HER2-positive. In contrast, the ZR-75-1 and MCF-7 cell lines that were HER2-negative and had a wild-type p53 gene exhibited a lack of response to hirsutine-induced cytotoxicity. The experimental results indicate that hirsutine triggered apoptosis in MDA-MB-453 cells, whereas this did not occur in MCF-7 cells. This effect was achieved through the stimulation of caspases. The study found that hirsutine had numerous implications on different types of breast cancer cells. This was evidenced by an increase in the expression of γH2AX. The study found that the MDA-MB-453 cells exhibited a decrease in the activity of the Akt, NF-κB, and HER2 pathways, and a stimulation of the p38 MAPK pathway. Additionally, a triggering of the DNA damage was observed. The study found that the expression of γH2AX, which is a well-known indicator of DNA damage response, was notably increased following the administration of hirsutine in conjunction with the p38/MAPK stress signaling pathways. This suggests that the hirsutine-associated apoptosis may have been triggered by a DNA damage response [47].

### 3.4. Fumigaclavine C

Fumigaclavine C was applied to MCF-7 cells in a range of doses to determine its effects on the suppression of proliferative activity and triggering of apoptosis in breast cancer cells. The analysis of anti-proliferation was conducted through an assessment of cell mobility and the signaling pathway of MAPK. To determine the specific mechanism via which fumigaclavine C inhibits the production of MMP, we examined the impact of fumigaclavine C on the ERK 1/2, JNK, and p38 MAPK signaling pathways. The MCF-7 cell line was treated with fumigaclavine C at different doses (20–60 μM). The treatment resulted in a dose-dependent inhibition of the phosphorylation of ERK 1/2, JNK, and p38. The findings indicated that the inhibitory impact of fumigaclavine C on cell proliferation was achieved by obstructing the signal transduction of specific molecules in the MAPK pathway, including ERK, JNK, and p38 MAPK signaling pathways, in triggered MCF-7 cells. Hence, it is evident that the potential molecular mechanism responsible for the inhibition of proteinases by fumigaclavine C is likely the suppression of the stimulation of these signaling pathways [48].

### 3.5. Flavopereirine

A study investigated the involvement of MAPK in autophagy caused by flavopereirine. It was shown that the expression of p-p38 MAPK, triggered by flavopereirine, was reduced in MDA-MB-231 cells treated with the p-AKT plasmid compared to cells treated with the control plasmid. The levels of both total and phosphorylated p38 MAPK were decreased after specifically inhibiting p38 MAPK using a targeted siRNA in the presence of flavopereirine. The increase in LC3-II produced by flavopereirine was also reduced with the downregulation of p38 MAPK. The data suggest that p-AKT acts as a regulator upstream of p-p38 MAPK, and it is responsible for the increase in LC3-II accumulation caused by flavopereirine. The compound flavopereirine has been observed to cause cell cycle arrest and activate the AKT/p38 pathway, both of which are implicated in the initiation of cell death in MDA-MB-231 cells [49].

Flavopereirine has been found to significantly reduce cellular viability in colorectal cancer cells, leading to both extrinsic and intrinsic apoptosis and promoting G2/M arrest. Flavopereirine inhibited signaling between JAKs-STATs and c-Myc in CRC cells but compelled expressions of constitutively active c-Myc and STAT3 were unsuccessful in reversing the loss in cell viability. Flavopereirine also increased the levels of phosphorylation of P53 in CRC cells, indicating that P53 activity plays a critical role in flavopereirine-induced cell growth regulation. Flavopereirine is a potential selective treatment for lung cancers that display functional p53 but not nonfunctional p53. However, the exact mechanism through which flavopereirine triggers apoptosis in mutant P53-expressing CRC cells is not yet fully understood. Flavopereirine activates P53 and P21, suppressing cell growth through extrinsic and intrinsic apoptosis pathways and arresting the G2/M-phase **[50]**.

### 3.6. Evodiagenine 

A study investigates the effects of evodiagenine controlled photodynamic therapy on MDA-MB-231, a type of extremely invasive breast cancer cell line, specifically focusing on its photo-cytotoxicity. After cancer cells have been treated with evodiagenine for a prolonged time period, they are subjected to UV light for an hour and a half. The combination of evodiagenine plus photodynamic treatment was shown to significantly decrease the likelihood of breast cancer cells surviving, in addition to boosting reactive oxygen species (ROS) generation and lactate dehydrogenase (LDH) release. A Western blot was conducted to find out the degree to which apoptosis-related proteins (AKT/PI3K/mTOR/p38 MAPK) had been phosphorylated. Without being irradiated, evodiagenine exhibited photochemical activity and exhibited a low level of toxicity against both normal cells and malignant cells. Evodiagenine-PDT induced a significant upregulation of p38 phosphorylation in MDA-MB-231 cells. The work indicates that evodiagenine-PDT has the ability to stimulate the p38 pathway, implying that p38 phosphorylation may play a role in the apoptosis produced by evodiagenine-PDT (Figure 2) [51].

### 3.7. 3,3′-Diindolylmethane 

Antiproliferative and apoptosis-inducing effects on gastric cancer are exhibited by 3,3′-diindolylmethane. These effects are believed to be mediated by the TRAF2-p38 MAPK axis. According to the results, 3,3′-diindolylmethane was observed to suppress the expressions of TRAF2 while activating p-p38 MAPK and the resulting downstream protein p-p53. These effects were found to be associated with DIM-induced proliferation, inhibition, and apoptosis induction in cells. The observed effects of 3,3′-diindolylmethane were counteracted by either overexpression of TRAF2 or by administering a specific inhibitor of p38 MAPK (SB203580). The data gathered indicates that the regulation of the TRAF2/p38 MAPK signaling pathway is crucial in preventing the growth of gastric cancer and promoting apoptosis through 3,3′-diindolylmethane. The study’s results enhance comprehension of how 3,3′-diindolylmethane operates as a modulator of TRAF2 from a pharmacological perspective. This discovery also presents a fresh therapeutic objective for treating gastric cancer in humans [52]. 

Zhu et al. reported that, on cervical cancer cells, 3,3′-diindolylmethane has appeared to have pro-apoptotic and anti-proliferative properties. SiHa cells displayed a higher degree of sensitivity than HeLa cells did in this regard. The 3,3′-diindolylmethane-induced pro-apoptotic responses in cervical cancer cells are associated with the MAPK and PI3K signaling pathways [53].

In the prostate cell line PC-3, siRNA suppression of p38 MAPK protein blocked p75^NTR^ activation by 3,3′-Diindolylmethane. Within 1 min of being exposed to 3,3′-diindolylmethane, p38 MAPK was phosphorylated. As a result, p75^NTR^-dependent apoptosis was induced by 3,3′-diindolylmethane in prostate cancer cells via the p38 MAPK pathway [54].

Vivar et al. demonstrated that in human prostate cancer cells, 3,3′-diindolylmethane may inhibit cell cycle progression by regulatory pathways in a distinct manner. This was found to be the case regardless of the androgen dependence of the cells and the status of P53. The action of 3,3′-diindolylmethane on the cell cycle in DU145 cells is mediated by the p38 MAPK and Sp1 pathways [55] (Table 1).

### 3.8. Idole-3-Carbinol 

As per the findings of a conducted experiment, the growth of hepatic stellate cells is inhibited by Indole-3-carbinol through the obstruction of the NADPH oxidase/ROS/p38 MAPK pathway. The findings suggest that I3C has the ability to effectively suppress the proliferation of HSCs concentration-dependently, regardless of the presence or absence of PDGF-BB stimulation, as evidenced by statistical significance (*p* < 0.01). I3C has the potential to impede the progression of hematopoietic stem cells (HSC) in the G(0)/G(1) phase by preventing their entry into the S phase. The levels of protein and mRNA expressing α-smooth muscle actin treated with I3C were observed to have a significant reduction (*p* < 0.01). The study found that I3C had a significant impact on various factors, including a decrease in cyclin D(1) and CDK4 mRNA expressions, NADPH activity, and ROS production. The individual’s observation indicates that I3C has a concentration-dependent inhibitory effect on the phosphorylated p38 MAPK in HSC-T6. However, the phosphorylated ERK1/2 MAPK remains unaffected by I3C. The research proposes that I3C is potent in hindering the growth of HSC by obstructing the NADPH oxidase/ROS/p38 MAPK pathway. The findings indicate that the consumption of I3C through diet could potentially have a beneficial impact on preventing and treating chronic liver diseases [56].

### 3.9. Notoamide G

Both the autophagy and apoptosis pathways were discovered to be responsible for Notoamide G’s capacity to limit Huh-7 and HepG2 cell viability. This suggests that notoamide G promoted apoptosis through a mitochondrial mechanism as well as a dead receptor-mediated pathway since it increased the production of caspase-3, -8, and -9 in conjunction with the devastation of the PARP gene dose-dependently. Not only that but after 24 h, notoamide G significantly upregulated the levels of the essential proteins LC3B and Beclin1, which led to a rise in the autophagic vacuole size in both Huh-7 and HepG2 cells dose-dependently. Further analysis found that notoamide G induced phosphorylation of JNK and P38, although its effect on the total protein of JNK and P38 was only marginal. Notoamide G was able to trigger autophagy as well as apoptosis at the same time, and both of these processes were facilitated through the stimulation of the P38/JNK pathway (Figure 2) [57]. 

### 3.10. Vinblastine 

Deacon et al. observed that vincristine (1 μM) or vinblastine (1 μM) treatment of HeLa cells induced mitotic arrest and activated p38 MAPK selectively in the mitotically arrested population. Vincristine and vinblastine were able to activate JNK more effectively in cells that were connected (Figure 2) [58].

### 3.11. Vincristine

The study conducted by Zhu and colleagues revealed that vincristine exhibited an upregulatory effect on the expression of Bax, Bak, PUMA, Noxa, p53 and p21 proteins, while concurrently downregulating and/or phosphorylating the Bcl-2 protein. The suppression of JNK, but not p38 mitogen-activated protein kinase, exhibited a noteworthy hindrance of vincristine-induced apoptosis in IgR3 and Mel-RM cells. Furthermore, the phosphorylation induced by vincristine and the subsequent reduction in Bcl-2 was found to be inhibited by a JNK inhibitor. The impact of RNA interference-induced downregulation of p53, PUMA, or Bim mRNA on vincristine-induced apoptosis in IgR3 cells was found to be negligible or insignificant [59]. 

### 3.12. Sclerotiamides C

Sclerotiamides C and D are classified as alkaloids of the notoamide type, which are distinguished by the presence of a distinctive 2,2-diaminopropane unit. Additionally, sclerotiamides E and F are novel notoamide hybrids that contain a previously unidentified coumarin unit. Sclerotiamide H denotes a novel notoamide framework that is highly oxidized. The compounds Sclerotiamides C and F exhibited noteworthy inhibition against a range of cancer cell lines. The compound known as Sclerotiamide C has been observed to trigger apoptosis in HeLa cells through a number of mechanisms, including cell cycle arrest, the activation of ROS production, and the regulation of apoptosis-related proteins within the MAPK pathway. This research expands the range of scaffolds present in notoamides and presents a promising candidate for the creation of a cytotoxic agent [60]. 

### 3.13. Ellipticine

Kim et al. reported that exposure to ellipticine at concentrations between 1–10 μM induced RL95-2 human endometrial cancer cells to apoptosis. The death of cells induced by ellipticine was linked with a rise in the proportion of cells in the cell cycle’s G2/M phase. During treatment with ellipticine, intracellular ROS was produced, which were then maintained at elevated levels. ERK and JNK were activated after ROS formation in ellipticine-treated cells (Figure 2). Caspases, ROS, and ERK all appeared to have an effect on the release of apoptosis-inducing factors from mitochondria [61]. 

### 3.14. Chaetoglobosin K

A research report explores the mechanisms of regulation that govern Chaetoglobosin K’s dual influence on the Akt and JNK signaling pathways. The report also illustrates how Chaetoglobosin K modifies the JNK pathway in ras-transformed and human lung cancer cells. According to the findings of this research, chaetoglobosin K is able to modulate the activation of JNK in both WB-ras1 and human lung carcinoma cells. According to the findings, Chaetoglobosin K was successful in inhibiting Akt and JNK phosphorylation at critical activation sites in both ras-transformed cells and human lung cancer cells. The consequence of this was an influence on the downstream effectors of both kinases. Chaetoglobosin K did not have any effect on the direct upstream kinases of JNK. Wortmannin and LY294002, which are both PI3 kinase inhibitors, decreased Akt phosphorylation in ras-transformed cells but did not influence JNK phosphorylation. This research indicates that Chaetoglobosin K has an inhibitory impact on both the Akt and JNK signaling pathways in ras-transformed epithelial cells as well as human cancer cells. The exclusive dual impact that Chaetoglobosin K has on these important pathways in the carcinogenesis process is what sets it apart. Chaetoglobosin K with the purpose of doing more research into determining its molecular target(s) and its potential anti-tumor activity in vivo [73].

In a study, it was shown that the effect of Jerantinine B on acute myeloid leukemia cells is oxidative stress-dependent, which activates c-Jun early. Jerantinine B caused early death in cells, and phospho-kinase arrays showed that total and phosphorylated c-Jun (S63) increased and activated. The necessity of active c-Jun/JNK for JB-induced apoptosis was verified to exist by the use of pharmacological suppression of MAPK/JNK. According to these results, the action of Jerantinines B (JB) on acute myeloid leukemia (AML) cells is subject to oxidative stress (OS), which serves as an immediate inducer for c-Jun stimulation. Furthermore, it was claimed that the primary biological targets are achieved through c-Jun/JNK signaling. In point of fact, several different compounds that have pro-oxidant capabilities were found to be beneficial against leukemic cell lines [63].

### 3.15. Jerantinine B

In a study, it was shown that the effect of JB on acute myeloid leukemia cells is OS-dependent, which activates c-Jun early. JB caused early death in cells, and phospho-kinase arrays showed that total and phosphorylated c-Jun (S63) increased and activated. The necessity of active c-Jun/JNK for JB-induced apoptosis was verified to exist by the use of pharmacological suppression of MAPK/JNK. According to these results, the action of JB on AML cells is subject to OS, which serves as an immediate inducer for c-Jun stimulation. Furthermore, it was claimed that the primary biological targets are achieved through c-Jun/JNK signaling. In point of fact, several different compounds that have pro-oxidant capabilities were found to be beneficial against leukemic cell lines [63].

### 3.16. Harmalacidine

Some of the other indole alkaloids were tested for how effectively they could kill human cells with leukemia, and the results indicated that some of them exhibited significant activity. Harmalacidine (HMC) showed the greatest level of cytotoxicity when tested against U-937 cells. PTKs-Ras/Raf/ERK were the targets of HMC’s cytotoxic mechanism in order to achieve this effect. The findings provided compelling evidence that the alkaloids derived from *Peganum harmala* have the potential to be a fruitful choice for the treatment of leukemia [64].

### 3.17. L20

It was found that the natural alkaloid known as calothrixin B exhibits minimal levels of cytotoxicity in leukemia cells. This property is attributed to the compound’s unique indolo[3,2-j] phenanthridine structure. Although L20, a new derivative of calothrixin B, possesses potent anti-proliferation properties, it is also capable of inducing mitochondria-mediated apoptosis and a block in the G2/M phase of the cell cycle. This can be accomplished through the induction of DNA damage and through the inhibition of the p38 MAPK pathway. These effects are accompanied by a decrease in the expression of p-ERK and c-Myc proteins in HEL cells. Anyway, L20 has promise for use as a potent chemotherapeutic medication in the management of erythrocytic leukemia [65,74].

### 3.18. 11-Methoxytabersonine

11-methoxytabersonine was found to kill lung cancer cells by producing necroptosis independent of apoptosis in a study [66]. The compound 11-methoxytabersonine effectively eliminated lung cancer cells by triggering necroptosis, a kind of cell death that is not reliant on apoptosis. Furthermore, 11-methoxytabersonine significantly stimulated autophagy in both cell lines, hence providing protection against 11-methoxytabersonine-induced necroptosis. Ultimately, the autophagy induced by 11-methoxytabersonine was discovered to occur via the activation of JNK signaling systems in both cells. Collectively, 11-methoxytabersonine displayed a distinct anti-tumor mechanism that differs from previously documented analogs. This molecule has the potential to be a primary candidate for the advancement of novel chemotherapeutic treatments for lung cancer [66].

### 3.19. Calothrixin A

CAA45 is a compound belonging to the calothrixin family that has anti-cancer action at concentrations as low as nanomolar. Through the suppression of Topo I, CAA45 was able to exercise its anti-lung cancer action. This resulted in the cell cycle arrest and the migration of cells, as well as the stimulation of mitochondrial cell death and autophagy through the PI3K/Akt/JNK/p53 pathway. At a concentration of 10 μM, CAA45 was able to suppress Topo I activity more effectively than CPT did under the same experimental circumstances. In addition to this, CAA45 caused a halt in the cell cycle during the S phase in A549 cells, which is an indication of DNA damage inside the cell. In light of the information presented above, we are able to draw the conclusion that CAA45’s inhibition of the Topo I enzyme was a contributing factor in the DNA damage that led to a halt S phase of the cell cycle and suppression of cell proliferation. The mitochondrial-induced apoptosis pathway was responsible for regulating the pro-apoptotic impact that CAA45 possessed. The Akt/JNK/p53 signaling pathway was responsible for regulating CAA45’s triggered apoptosis and autophagy in cells [67].

### 3.20. Harmol

In human non-small cell lung cancer (NSCLC) H596 cells, Harmol was shown to promote apoptosis by activating caspase-8 in a manner that was independent of the interaction between Fas and Fas ligands. In human NSCLC A549 cells, harmol causes autophagy, which ultimately leads to cell death. Harmol did not stimulate caspase-3, -8, or -9 function in A549 cells despite the fact that it caused apoptosis in a substantial dose- and time-dependently. In addition, treatment with harmol did not result in the breakdown of poly-(ADP-ribose)-polymerase in A549 cells. In A549 cells that had been subjected to 70 mM harmol, an electron microscope revealed that autophagy was present, but apoptosis was not. Although medical care with harmol had no apparent effect on the Akt/mTOR pathway, it could subsequently stimulate the ERK1/2 pathway. However, suppression of autophagy was achieved by inhibiting the ERK1/2 pathway using the MEK/ERK inhibitor U0126. This resulted in a partial reduction in autophagy. Because of this, the triggering of the ERK1/2 pathway may be associated with harmol-mediated autophagy; nevertheless, it is possible that another significant mechanism has been found in A549 cells. The induction of autophagy by harmol exhibits a correlation with the ERK pathway, although with partial dependency. Hence, it is possible that an additional basic way is implicated in the autophagy initiation mechanism of harmol [68].

### 3.21. Chaetoglobosin G

The cell growth of A549 cells was observed to be inhibited dose-dependently upon exposure to Chaetoglobosin G. The analysis of transcriptome sequencing revealed that Chaetoglobosin G significantly caused cell cycle arrest. The experimental results obtained from flow cytometry analysis and western blot analysis demonstrated that Chaetoglobosin G elicited G2/M arrest in the cell cycle. Additionally, the protein expression levels of p21 were found to be upregulated, while those of cyclinB1 were downregulated. The administration of Chaetoglobosin G leads to a substantial reduction in protein expression of EGFR, p-EGFR, p-ERK, and p-MEK. The findings indicate that the induction of the autophagy process by Chaetoglobosin G is achieved through the inhibition of the EGFR/ERK/MEK/LC3 signaling pathway [69].

### 3.22. 3α-Acetonyltabersonine

The death of glioblastoma stem cells may be induced by 3α-acetonyltabersonine, which has the potential to inhibit the growth of tumor cells. In addition, a comprehensive assessment of transcriptome research and Western blot analysis suggested that the MAPK pathway became activated by phosphorylation after being treated with 3α-acetonyltabersonine. The accumulation of DNA damage increased activation of the MAPK pathway, which ultimately led to the induction of apoptosis. The 3′-acetonyltabersonine was the agent that impeded the repair of DNA damage mechanisms. It is of crucial importance and enhances the anticancer mechanism in indole alkaloids to have 3α-acetonyltabersonine, which prevents DNA damage repair and the buildup of DNA damage that impacts the genome integrity [70].

### 3.23. Isomahanine

The CLS-354/DX cellular model, which exhibits an overexpression of multidrug resistance-associated protein 1 (MRP1), demonstrated resistance towards the anticancer agent’s cisplatin and camptothecin. The cytotoxicity induced by isomahanine was observed to be effective against CLS-354/DX cells, irrespective of their resistance status. The induction of apoptosis was evaluated following treatment with isomahanine. The results indicated a noteworthy time-dependent increase in apoptosis. The induction of caspase-dependent apoptosis by isomahanine was assessed through the utilization of z-VAD fmk. Isomahanine induced the activation of protein kinase PNA (PKR)-like ER kinase (PERK), which serves as an endoplasmic reticulum stress sensor time-dependently. Simultaneously, the aforementioned compound elicited the upregulation of the transcription factor CHOP, which serves as an indicator of the unfolded protein response (UPR). Following a 6-h exposure to isomahanine, there was a considerable rise in the phosphorylation of JNK1/2p38, and ERK1/2. After 12 h of treatment, there was a notable increase in the phosphorylation of p38 and ERK1/2, whereas their activation exhibited a slight decline after 24 h. The activation of JNK1/2 reached its maximum level at 6 h and subsequently declined after 12 h. The compound known as Isomahanine has demonstrated the ability to elicit endoplasmic reticulum stress, which has the potential to modulate both apoptosis and autophagic cell death pathways through the activation of p38 MAPK [71].

### 3.24. Dehydrocrenatidine

The apoptosis in human oral cancer cells by dehydrocrenatidine was observed and the induction of apoptosis by dehydrocrenatidine was facilitated by its stimulation of ERK and c-JNK, as evidenced by the cotreatment of dehydrocrenatidine with MAPK inhibitors. The restricted levels of dehydrocrenatidine could potentially impede the modulation of p-AKT expression by dehydrocrenatidine. The current research has shown that dehydrocrenatidine induces apoptosis in cells through the activation of several proteins, including AKT, ERK1/2, JNK1/2, and p38 MAPK. However, further research is necessary in order to identify whether or not the MAPK pathway is regulated in a directed approach [72].

## 4. Clinical Trials

There is a large family of therapeutically active substances known as indole alkaloids. Many of these substances are nowadays used clinically across a variety of disciplines [75]. Till now, some clinical investigations have been carried out to assess the potential of indole alkaloids in cancers that have already been proven as potential candidates preclinically. Vinorelbine is a drug that is extensively researched and regularly used for NSCLC, either as a monotherapy or in conjunction with other treatments. It belongs to the group of four primary vinca alkaloid drugs, which include vinflunine, vinblastine, vinorelbine, and vincristine. Vinorelbine, a semi-synthetic derivative of vinblastine, has received FDA approval to treat malignant patients of lung cancer in the United States [76]. A Phase III clinical trial showed that the combination of vinorelbine and cisplatin showed a 35% annual survival rate, outperforming the average yearly survival rate for vinorelbine alone at 30%. The combination therapy also outperformed the ifosfamide and epirubicin combination therapy, with response rates of 21% and 47%, respectively [77,78]. According to a randomized trial conducted by Le Chevalier et al., the combined administration of vinorelbine and cisplatin resulted in longer life expectancy and a higher response rate compared to the combination treatment of vindesine and cisplatin or the use of vinorelbine alone. The study also reported acceptable levels of toxicity among the 612 patients involved [79]. Tabchi et al.’s study on 107 patients with Stage III NSCLC found that the vinblastine-cisplatin combination treatment was more effective than the paclitaxel-carboplatin combination and caused less harm than the etoposide-cisplatin combination, based on survival rates and toxic effects analysis [80]. Spigel et al. performed a Phase II clinical trial which possessed 51 patients. The results showed that vinflunine produced partial responses in 17% and 20% of patients with relapse-refractory SCLC and relapsed SCLC, respectively. This represented a much larger prevalence than was projected for this group of patients. Unfortunately, regardless of how high the average response rate was, the overall study group did not see an improvement in their chances of survival [81]. Talbot et al. carried out a Phase II study with the objective of investigating the safety and efficacy of primary vinflunine therapy to treat malignant pleural mesothelioma (MPM). Vinflunine exhibited efficacy in certain patients diagnosed with MPM, as evidenced by 10.8 months of overall survival and a 13.8% response rate [82].

Patients with metastatic triple-negative breast cancer were the subjects of a Phase II investigation of the combination of chidamide and cisplatin that was being conducted in a clinical trial. This particular research included a total of sixteen different patients. There were a total of these, and fifteen were accessible for analysis. Four of these fifteen individuals showed established signs of having objectively positive responses. Because the inclusion of chidamide did not result in an increase in the effectiveness of cisplatin in the first-line therapy against advanced triple-negative breast cancer (TNBC), the phase II clinical study was terminated before it could continue. In what seems to be the first trial of its kind to evaluate the effects of a histone deacetylase inhibitor in TNBC patients, our research produced unsatisfactory findings that need to guide further research [83]. A Phase I research study involved 79 patients with severe malignancy treated with PSC 833, an infusional drug. The study predicted pharmacokinetic interactions between PSC 833 and vinblastine, leading to decreased vinblastine dosages when toxicities were found, and increased PSC 833 doses. The study used three different regimens of PSC 833 and two different formulations, examining the effects of these interactions on cancer treatment [84].

## 5. Toxicity and Adverse Effects

Vincristine is a significant vinca alkaloid that is extensively used in front-line combination chemotherapy therapies for cancer. Because of its ability to lower testosterone levels, vincristine contributes to reproductive damage [85]. In addition, Sonawane and colleagues discovered that therapy with vincristine promotes epididymal toxicity, which is another factor that adds to the total reproductive toxicity. In one set of experiments, mature male Wistar rats that had previously been shown to be fertile were given vincristine by intraperitoneal injection at a level of 40 mg/kg each day for a period of 30 days. According to the findings, the injection of vincristine was likely responsible for aberrant sperm numbers as well as decreased function, both of which led to infertility in the end. Vincristine is known to have a number of negative side effects in addition to its impact on reproduction, including hyponatremia, hair loss, constipation, and peripheral neuropathy [86]. Vinblastine, a vinca alkaloid isolated from Catharanthus roseus, has antineoplastic activity but can cause severe allergic reactions, bone marrow toxicity, urine blood, infection, bleeding, bone pain, shortness of breath, constipation, headache, vomiting, stomach pain, loss of appetite, and deep ulcers [86]. Vinblastine has been linked with the development of deep ulcers. According to the findings of Harding and colleagues, dogs who were given vinblastine at a dosage of 2 mg/m^2^ had symptoms of gastrointestinal toxicity, which are often linked with hyporexia and diarrhea [86,87].

It has been shown that evodiamine may lower the survival of L-02 cells while simultaneously increasing the activity of aspartate transaminase, ALT, LDH, and ALP. This might be because evodiamine activates p38 beta. P38 inhibitors could be able to control this phenomenon [88,89]. Conversely, research investigating cardiotoxicity discovered that evodiamine exhibited a 50% inhibitory concentration in a laboratory setting. It also resulted in increased release of LDH and malate dialdehyde, while decreasing the activity of superoxide dismutase. When the concentration of zebrafish was raised to 1600 ng/mL in living organisms, it resulted in 100% mortality. This rise in concentration led to cardiac insufficiency, alterations in heart rate and circulation, as well as deformation of the pericardium [89,90].

## 6. Conclusions and Future Perspective

Preclinical research investigations support the crucial role of indole alkaloids in modulating MAPK pathways in cancers. Indole alkaloids from different sources have the ability to alter the MAPK pathway and reduce cell growth, inhibiting the cell cycle, and preventing angiogenesis. ERK, JNK, and p38 MAPK are associated with several malignancies. The in vitro experiments have demonstrated their significance in the management of breast, gastric, colon, hepatocellular, pancreatic, cervical, endometrial, ovarian, leukemia, lungs, skin, bladder, prostate, neural, and oral cancers. Evodiamine is a significant indole alkaloid in cancer treatment. Also, some clinically trialed indoles such as vinblastine, vincristine, vinorelbine, and vinflunine are also spot-highlighted and gained research interests for further clinical safety. Further clinical tests are necessary to verify the anticancer potential of indole alkaloids via MAPK signaling pathways. Furthermore, there are no significant toxicological studies reported till now. It is necessary to evaluate the toxicological profile of indole alkaloids to achieve the desired therapeutic goal.

## Figures and Tables

**Figure 1 cancers-15-05311-f001:**
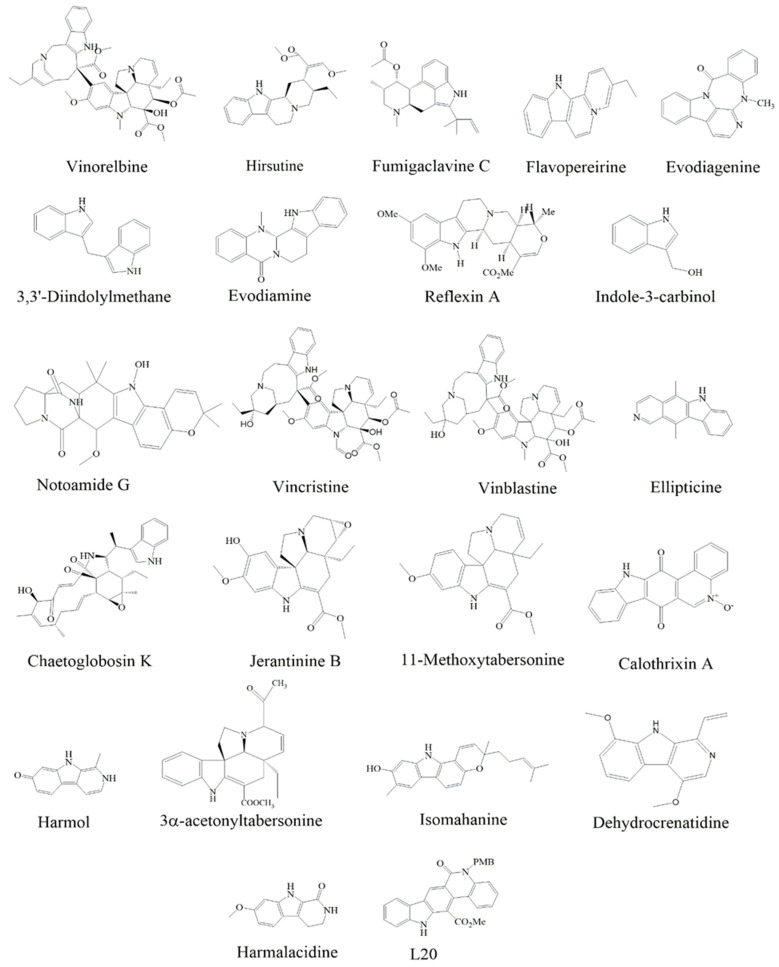
Structures of anticancer indole alkaloids associated with MAPK pathway.

**Figure 2 cancers-15-05311-f002:**
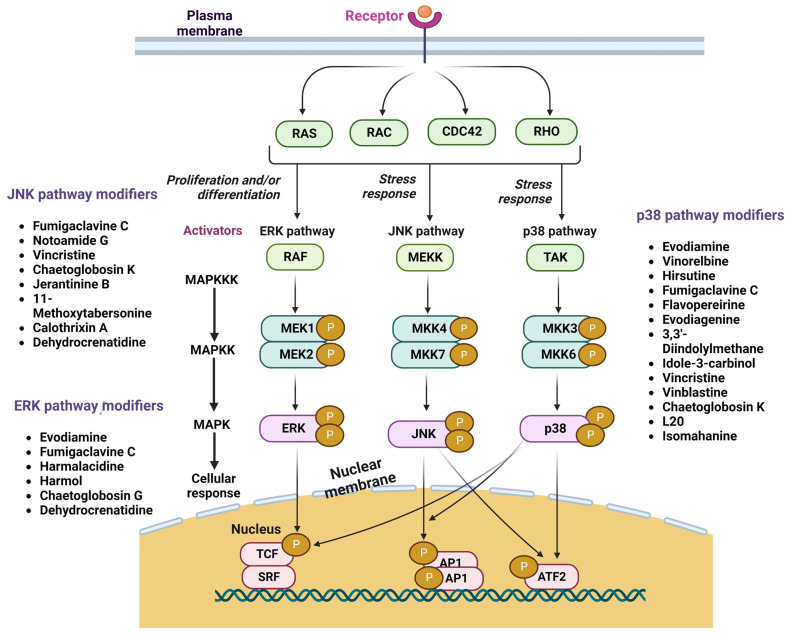
Illustration represents MAPK signaling pathways and Indole alkaloids as its modifiers.

**Table 1 cancers-15-05311-t001:** Preclinical findings on the use of indole alkaloids in cancers.

Compounds	Cancer Type	Findings	Ref.
Evodiamine	Breast cancer	The p38 MAPK and ERK pathways are controlled by evodiamine.	[37]
	Colon cancer	JNK activation induces apoptosis triggered by evodiamine	[39]
	Pancreatic cancer	Evodiamine dose-dependently stopped AKT, ERK1/2, and P38-MAPK phosphorylation.	[40]
	Ovarian cancer	When evodiamine is present, PI3K/Akt, p38 MAPK, and ERK1/2 MAPK activation may result in cell death.	[41]
	Skin cancer	Evodiamine induced caspase-mediated apoptosis and necrosis through the activation of p38 and ERK MAPK.	[42]
	Bladder cancer	When evodiamine was added to 5637 and HT1197 cells, the P38-MAPK and some of the JNK pathways became more active while ERK pathway only became less active in 5637 cells.	[43]
	Neural cancer	Glioma cells are destroyed by evodiamine via autophagy triggered by intracellular calcium/JNK signaling and apoptosis triggered by calcium/mitochondria.	[44]
	Lung cacner	p-ERK and ERK-MAPK expression was inhibited by evodiamine.	[45]
Vinorelbine	Breast cancer	Vinorelbine increased P38-MAPK activity in breast cancer cells MCF-7 and MDA-MB-468.	[46]
Hirsutine	Breast cancer	In addition to activating the p38-MAPK pathway, DNA damage signaling was induced in MDA-MB-453 cells	[47]
Fumigaclavine C	Breast cancer	Fumigaclavine C dose-dependently reduced JNK, ERK 1/2, and P38-MAPK phosphorylation.	[48]
Flavopereirine	Breast cancer	Flavopereirine causes MDA-MB-231 cell death via cell cycle arrest and AKT/P38-MAPK signaling pathway.	[49]
	Colon cancer	Flavopereirine elicited a decrease in cellular viability, impeded progression of the cell cycle, and prompted apoptosis via P53 signaling.	[50]
Evodiagenine	Breast cancer	Evodiagenine may inhibit PI3K/AKT/mTOR and increase P38-MAPK phosphorylation to cause photocytotoxicity	[51]
3,3′-Diindolylmethane	Gastric cancer	3,3′-diindolylmethane inhibits gastric cancer proliferation and triggers apoptosis via altering the TRAF2/p38-MAPK signaling pathway.	[52]
	Cervical cancer	The MAPK and PI3K signaling pathways have been shown to be involved in 3,3′-Diindolylmethane’s pro-apoptotic actions on cervical cancer cells.	[53]
	Prostate cancer	3,3′-Diindolylmethane as an indole capable of eliciting p75^NTR^-dependent apoptosis in prostate cancer cells via the p38-MAPK pathway.	[54]
	Prostate cancer	In DU145 cells, 3,3′-diindolylmethane activates the p38-MAPK pathway in order to activate p27Kip1 and stop the cell cycle in G1.	[55]
Idole-3-carbinol	Hepatocellular cancer	Indole-3-carbinol may inhibit the hepatic stellate cell growth by blocking the NADPH oxidase/ROS/p38-MAPK pathway.	[56]
Notoamide G	Hepatocellular cancer	Notoamide G triggered a P38/JNK pathway, which resulted in the induction of apoptosis and autophagy	[57]
Vincristine	Cervical cancer	Vincristine caused a mitotic arrest, and only the cells that were experiencing a mitotic arrest had p38-MAPK active.	[58]
	Skin cancer	Apoptosis produced by vincristine is JNK/MAPK activation dependent in at least some melanoma cell lines.	[59]
Vinblastine	Cervical cancer	Vinblastine induced mitotic arrest and selectively activated p38-MAPK in mitotically arrested cells.	[58]
Sclerotiamides C	Cervical cancer	Sclerotiamide C has been observed to prompt apoptosis in HeLa cells through the mechanisms of cell cycle arrest, ROS production activation, and regulation of proteins related to apoptosis within the MAPK signaling pathway.	[60]
Ellipticine	Endometrial cancer	In RL95-2 cells, apoptosis can be induced by ellipticine through ROS and the MAPKs activation	[61]
Chaetoglobosin K	Ovarian cancer	Chaetoglobosin K increased P38-MAPK phosphorylation in ovarian cancer cells, causing G2 arrest through cyclin B1 and death	[62]
Jerantinine B	Leukemia	Jerantinine B caused cell death, and phospho-kinase arrays revealed increased and activated total and phosphorylated c-Jun/JNK (S63) levels.	[63]
Harmalacidine	Leukemia	Harmalacidine’s cytotoxic action targeted mitochondrial and PTKs-Ras/Raf/ERK pathways.	[64]
L20	Leukemia	Through damaging DNA and p38-MAPK pathway modification, L20 caused mitochondrial driven apoptosis and G2/M arrest	[65]
11-Methoxytabersonine	Lung cancer	The autophagy induced by 11-methoxytabersonine was discovered to occur through the activation of JNK signaling systems.	[66]
Calothrixin A	Lung cancer	Calothrixin A acted against lung cancer by blocking Topo I. This caused the cell cycle to stop and the cells to move, as well as apoptosis and autophagy through the PI3K/Akt/JNK/p53 pathway.	[67]
Harmol	Lung cancer	MEK/ERK inhibitor U0126 stopped autophagy in part by blocking the ERK1/2 pathway.	[68]
Chaetoglobosin G	Lung cancer	Chaetoglobosin G clearly stopped the growth of A549 cells, and it may have done this by causing autophagy in A549 cells via the ERK pathway to increase the production of P21.	[69]
3α-acetonyltabersonine	Neural cancer	3α-acetonyltabersonine hindered DNA damage repair, causing MAPK pathway activation and apoptosis	[70]
Isomahanine	Oral cancer	Isomahanine has been found to elicit endoplasmic reticulum stress, leading to the activation of both p38 MAPK-mediated apoptosis and autophagy	[71]
Dehydrocrenatidine	Oral cacner	The induction of apoptosis by Dehydrocrenatidine was observed through the stimulation of ERK and c-JNK.	[72]

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
