# Peer review of "Research Progress of Indole Alkaloids: Targeting MAP Kinase Signaling Pathways in Cancer Treatment"

_cancers, 2023, doi:10.3390/cancers15225311_

Round 1

Reviewer 1 Report

Comments and Suggestions for Authors

In their review article “Research Progress of Indole Alkaloids: Targeting MAP Kinase Signaling Pathways in Cancer Treatment” Al Amin and co-authors present very extensively numerous studies and results on the role of these compounds in the treatment of various cancer types. Unfortunately, I think that the description of all those other studies is too extensive and the whole article sounds like a retelling of other people’s works, which is not the aim of a review as the focus is completely lost. All in all, I find this review unreadable. That is why I cannot recommend it for publication in Cancers. Although I believe that the information in the review is valuable, I think that it must be completely re-structured and re-written.

1.       The organization of the review with separate paragraphs for every type of cancer is very cumbersome. I’d rather organize it with “chapters” about every alkaloid and its role in different cancer types.

2.       The information is not presented in a clear way. As I mentioned before it sounds like a retelling of works in too much detail so that one gets confused about the connection to the MAPK signaling pathway, which is supposed to be the focus of the review.

Examples:

-          The paragraph about gastric cancer basically illustrates the results of a single study (Ye et al. 2021).

-          In the paragraph about colon cancer the results by Fadaeinasab are retold in an unnecessary detailed way with IC50 values of experiments.

-          The paragraph about lung cancer also thoroughly retells the results of 3-4 studies with a lot of details about caspases, which is very confusing.

The examples I mentioned are just to illustrate my point of view, but this whole very long review is written like this. I think the authors do not really comprehend the aim of a review – to extract information from research papers, not just summarize it but analyze it in respect to their review’s focus and thesis. To me it is not clear what these are and as a potential reader of this work I am not able to find anything useful in this review except for the list of references.

3.       The Introduction contains too much general information about cancer and healthy living, which is out of place in a review that claims to be focused on molecular mechanisms of drugs. The last paragraph sounds bizarre – after giving basic information on cancer and then discussing indole alkaloids, the authors conclude “Therefore, the ultimate goal is to explore the role of MAPK signaling pathway in cancer progression.”

4.       It is not clear how the figures in the article contribute to anything. Figure 2 illustrates signaling pathways with the names of some indole alkaloids written next to proteins in these pathways without any indication how exactly they affect these proteins. Besides, the review is supposedly focused on MAPK signaling. Why are there two boxes/panels in Figure 2? The same critics applies to Figure 3.

5.       Table 1 is too extensive, and I believe that it should be reorganized in the same way I suggested for the main text – according to compound rather than cancer type. In the Findings of the table the focus should be on MAPK signaling and not any findings in general. I would remove the columns Dose and Study model, considering that all study models are cell cultures.

6.       I find all these statistics, with which each paragraph starts, unnecessary.

Author Response

Authors Response to Reviewer-1 Comments and Suggestions

Manuscript title: Progress of Indole Alkaloids: Targeting MAP Kinase Signaling Pathways in Cancer Treatment

Manuscript ID: cancers-2668470

Dear Editor,

Thank you for providing the evaluation of our manuscript. We have addressed the concerns and suggestion of all reviewers on a point-by-point basis as mentioned below. All the modifications in the revised manuscript have been indicated as yellow colored texts.

Dear Reviewer-1,

Many thanks for your efforts for the evaluation of our manuscript. We strongly believe that your valuable observations, concerns and suggestions will definitely improve the quality of our manuscript. As per your comments, we have addressed all the issues to the point. Please find our response against each comment. All the modifications in the revised manuscript have been indicated as yellow colored texts.

Reviewer Comments and Response:

#Reviewer-01:

Their review article “Research Progress of Indole Alkaloids: Targeting MAP Kinase Signaling Pathways in Cancer Treatment” Al Amin and co-authors present very extensively numerous studies and results on the role of these compounds in the treatment of various cancer types. Unfortunately, I think that the description of all those other studies is too extensive and the whole article sounds like a retelling of other people’s works, which is not the aim of a review as the focus is completely lost. All in all, I find this review unreadable. That is why I cannot recommend it for publication in Cancers. Although I believe that the information in the review is valuable, I think that it must be completely re-structured and re-written.

Response: Thank you very much for your criticism about the infrastructure of this manuscript. We have completely reorganized the manuscript as per your suggestion. Hope, you will value our effort.

  1. The organization of the review with separate paragraphs for every type of cancer is very cumbersome. I’d rather organize it with “chapters” about every alkaloid and its role in different cancer types.

Response: Thank you very much for your valuable suggestion to improve this manuscript. We have organized the manuscript with “chapters” about every alkaloid and its role in different cancer types.

  1. The information is not presented in a clear way. As I mentioned before it sounds like a retelling of works in too much detail so that one gets confused about the connection to the MAPK signaling pathway, which is supposed to be the focus of the review.

Response: Thank you very much for your critical observation. We apologize for the inconvenience. We have removed unnecessary information and focused on MAPK pathways and it’s subfamilies like ERK, JNK, and P38.

Examples:

- The paragraph about gastric cancer basically illustrates the results of a single study (Ye et al. 2021).

Response: Thank you. We have restructured the manuscript and reported alkaloids in headings, thus described their role on different cancer types.

- In the paragraph about colon cancer the results by Fadaeinasab are retold in an unnecessary detailed way with IC50 values of experiments.

Response: Thank you. We have removed unnecessary information except MAPK associated data.

- The paragraph about lung cancer also thoroughly retells the results of 3-4 studies with a lot of details about caspases, which is very confusing.

Response: Thank you very much for your deep insights into this manuscript. We have revised it accordingly.

The examples I mentioned are just to illustrate my point of view, but this whole very long review is written like this. I think the authors do not really comprehend the aim of a review – to extract information from research papers, not just summarize it but analyze it in respect to their review’s focus and thesis. To me it is not clear what these are and as a potential reader of this work I am not able to find anything useful in this review except for the list of references.

Response: Thank you. We have revised the whole manuscript as per your valuable suggestions. We hope, you will value our effort.

  1. The Introduction contains too much general information about cancer and healthy living, which is out of place in a review that claims to be focused on molecular mechanisms of drugs. The last paragraph sounds bizarre – after giving basic information on cancer and then discussing indole alkaloids, the authors conclude “Therefore, the ultimate goal is to explore the role of MAPK signaling pathway in cancer progression.”

Response: Thank you. We have revised the introduction section and carefully focused on MAPK pathways and its subfamilies.

  1. It is not clear how the figures in the article contribute to anything. Figure 2 illustrates signaling pathways with the names of some indole alkaloids written next to proteins in these pathways without any indication how exactly they affect these proteins. Besides, the review is supposedly focused on MAPK signaling. Why are there two boxes/panels in Figure 2? The same critics apply to Figure 3.

Response: We have removed the previous figure that are in question and added a new one focusing on the role of indole alkaloids in MAPK pathways and its subfamilies.

  1. Table 1 is too extensive, and I believe that it should be reorganized in the same way I suggested for the main text – according to compound rather than cancer type. In the Findings of the table the focus should be on MAPK signaling and not any findings in general. I would remove the columns Dose and Study model, considering that all study models are cell cultures.

Response: Thank you very much. We have deleted dose and study model column.

  1. I find all these statistics, with which each paragraph starts, unnecessary.

Response: Thank you for your valuable suggestions, and we have deleted all these accordingly.

Reviewer 2 Report

Comments and Suggestions for Authors

Journal of Cancers

Review Article;

The article entitled “Research Progress of Indole Alkaloids: Targeting MAP Kinase Signaling Pathways in Cancer Treatment’’. The authors have studied in wonderful way the progress of indol alkaloids in cancer treatment, as Cancer is the leading cause of morbidity and mortality of throughout the world. There are many signaling pathways involved in cancerous diseases, from which Mitogen-activated protein kinase (MAPK) pathway performs a significant role in this regard. Apoptosis and proliferation are correlated with MAPK signaling pathways. As Indole alkaloids have the anticancer potential through different pathways. Much more research is ongoing or completed with molecules belonging to this alkaloidal group. This study evaluate how indole alkaloids affect the MAPK signaling pathway in cancer treatment.

Comments for Authors

Ø  Write the keywords in alphabetical order.

Ø  The author needs to include the latest reference in the introduction section.

Ø  Could the author explain why he just focus in MAPK pathway?

Ø  The author needs to through light on the clinical trials in detail and can include the adverse effect of indole alkaloids.

Cite the following references;

·         DOI: 10.2174/1871520622666220831124321

Author Response

Authors Response to Reviewer-2 Comments and Suggestions

Manuscript title: Progress of Indole Alkaloids: Targeting MAP Kinase Signaling Pathways in Cancer Treatment

Manuscript ID: cancers-2668470

Dear Editor,

Thank you for providing the evaluation of our manuscript. We have addressed the concerns and suggestion of all reviewers on a point-by-point basis as mentioned below. All the modifications in the revised manuscript have been indicated as yellow colored texts.

Dear Reviewer-2,

Many thanks for your efforts for the evaluation of our manuscript. We strongly believe that your valuable observations, concerns and suggestions will definitely improve the quality of our manuscript. As per your comments, we have addressed all the issues to the point. Please find our response against each comment. All the modifications in the revised manuscript have been indicated as yellow colored texts.

Reviewer Comments and Response:

#Reviewer-02:

The article entitled “Research Progress of Indole Alkaloids: Targeting MAP Kinase Signaling Pathways in Cancer Treatment’’. The authors have studied in wonderful way the progress of indol alkaloids in cancer treatment, as Cancer is the leading cause of morbidity and mortality of throughout the world. There are many signaling pathways involved in cancerous diseases, from which Mitogen-activated protein kinase (MAPK) pathway performs a significant role in this regard. Apoptosis and proliferation are correlated with MAPK signaling pathways. As Indole alkaloids have the anticancer potential through different pathways. Much more research is ongoing or completed with molecules belonging to this alkaloidal group. This study evaluates how indole alkaloids affect the MAPK signaling pathway in cancer treatment.

Comments for Authors

Ø  Write the keywords in alphabetical order.

Response: Thank you very much for your suggestion. We have revised it accordingly.

Ø  The author needs to include the latest reference in the introduction section.

Response: Thank you very much for your suggestion. We have revised introduction section and included latest reference accordingly.

Ø  Could the author explain why he just focus in MAPK pathway?

Response: Thank you very much for your suggestion. Anticancer medications that target the MAPK signaling pathway provide an exciting new avenue for drug development. So we focused on this pathway.

Ø  The author needs to through light on the clinical trials in detail and can include the adverse effect of indole alkaloids.

Response: Thank you very much for your suggestion. We have revised clinical trial section and included adverse effect of indole alkaloids (Section 4).

Cite the following references;

DOI: 10.2174/1871520622666220831124321

Response: Thank you very much for your suggestion. We have cited the references (Ref. 18).

Round 2

Reviewer 1 Report

Comments and Suggestions for Authors

I appreciate the efforts of the authors concerning the restructuring of the review with chapters about each indole alkaloid compound. I also find that the figure and the table have been greatly improved. Still, I find it awkward that there are very short chapters of only several lines, e.g. the ones about vinblastine, vincristine, dehydrocrenatidine. Maybe the less studied substances should be separated in a chapter "Other indole alkaloids" and the others can be discussed more thoroughly. However, despite the improvement of the manuscript, my main concern has not been addressed by the authors - the review retells the works of other people without clear conclusions. As I pointed out in my previous review, it sounds more like a summary of other papers than a real review that points at major findings in the field that support a certain idea. Thus, I cannot recommend it for publication in Cancers.